# From Basic Research to Clinical Practice: Considerations for Treatment Drugs for Silicosis

**DOI:** 10.3390/ijms24098333

**Published:** 2023-05-05

**Authors:** Rou Li, Huimin Kang, Shi Chen

**Affiliations:** Key Laboratory of Molecular Epidemiology of Hunan Province, Hunan Normal University, Changsha 410013, China

**Keywords:** silicosis, IPF, pulmonary fibrosis, treatment drugs

## Abstract

Silicosis, characterized by irreversible pulmonary fibrosis, remains a major global public health problem. Nowadays, cumulative studies are focusing on elucidating the pathogenesis of silicosis in order to identify preventive or therapeutic antifibrotic agents. However, the existing research on the mechanism of silica-dust-induced pulmonary fibrosis is only the tip of the iceberg and lags far behind clinical needs. Idiopathic pulmonary fibrosis (IPF), as a pulmonary fibrosis disease, also has the same problem. In this study, we examined the relationship between silicosis and IPF from the perspective of their pathogenesis and fibrotic characteristics, further discussing current drug research and limitations of clinical application in silicosis. Overall, this review provided novel insights for clinical treatment of silicosis with the hope of bridging the gap between research and practice in silicosis.

## 1. Introduction

### 1.1. Silicosis

The prevalence of crystalline silicon dioxide dust is widespread in many areas [1]. As a result of the continuous inhalation of silica particles, many workers develop silicosis, an irreversible and incurable disease [2]. Silicosis is a chronic interstitial lung disease characterized by fibrosis, inflammation, and destruction of the pulmonary structures. It causes pulmonary hypertension, progressive dyspnoea and death from respiratory insufficiency [3]. Many measures have been taken in recent decades to protect workers in the workplace, but millions of workers still suffer from silicosis [4].

Silicosis is a severe concern in construction and mining workers, particularly young workers, who are exposed to quartz conglomerates during sandblasting, bolting, cutting, shaping, and installing kitchen countertops [5,6,7]. Recent studies have also indicated that exposure to nanosilica can cause inflammation and fibrosis in the lungs. The risk of nanosilica exposure in this emerging industry is noteworthy, despite the lack of reported cases [8]. The increasing number of silicosis cases worldwide presents new challenges for prevention in many countries [4]. The present circumstance underscores the significance of exercising caution in authorizing development of emerging industries and implementing early identification and control measures from a public health perspective. Furthermore, it is crucial to accelerate the discovery of remedies for silicosis from a clinical treatment standpoint.

### 1.2. Animal Models of Silicosis

Mice and rats have become popular choices for experimentation due to their diminutive size, ease of breeding and maintenance in large numbers, rapid life cycle and convenient sample collection. Among the commonly used strains, C57BL/6 mice and Wistar rats are frequently employed [9]. Interestingly, no overall difference was found in progressive fibrosis between female and male mice with silicosis [10]. Notably, the most effective volume of silica suspension instilled into the noses of inbred mice C57BL/6J, which are highly susceptible to silicosis, was 80 μL via repeated inhalation. After repeated exposure to 20 mg/mL, this mode of inhalation resulted in a high rate of successful entry into the lungs and a high survival rate [11].

Replicating and simulating the entire process of silicosis resulting from prolonged exposure to industrial dust poses significant challenges due to the physiological limitations of animals. Additionally, the combined toxic consequences of silica and any mineral, metal or bacterial material on its surface cannot be assessed in animal models compared to the actual inhalation of silicosis patients [12]. Both silica aerosols and suspensions must be disinfected prior to modeling, and the environments for establishing animal models of silicosis are different from the reality of workers inhaling silica particles, so there may be significant limitations between basic experiments and clinical drugs in the treatment of silicosis. Therefore, exploring and establishing a dynamic animal model highly similar to human silica inhalation is also a key direction for future research.

### 1.3. Pathogenesis of Silicosis

The pathogenesis of silicosis is not fully understood, and the disease is complex [12]. Although further research is required to clarify the role of intricate signaling pathways [13], multiple pathways are thought to be involved in the development of silicosis (Figure 1). Silica-induced lung injury is characterized by various mechanisms, including direct cytotoxic effects on macrophages, activation of macrophage surface receptors, lysosomal rupture, reactive oxygen species (ROS) production, inflammasome activation, cytokine and chemokine production, apoptosis/softening, and lung fibrosis [14].

## 2. Characteristics of Pulmonary Fibrosis

It is generally accepted that the pathogenesis of pulmonary fibrosis is characterized by persistent microdamage to the alveolar epithelium with an abnormal repair process. This process is characterized by abnormal activation of myofibroblasts, excessive accumulation of ECM, and lung scar formation, ultimately leading to structural destruction and loss of function in the lungs. Apart from silicosis, idiopathic pulmonary fibrosis (IPF) is also a prevalent disease that results in pulmonary fibrosis [15].

### 2.1. Characteristics of IPF

The origin of IPF, much like silicosis, remains unclear, with a convoluted pathogenesis likely involving multiple interconnected signaling pathways, including the TGF-β/Smad signaling pathway, Wnt/β-catenin signaling pathway, platelet-derived growth factor (PDGF) signaling pathway, PI3K/AKT signaling pathway and other signaling pathways [16]. IPF is thought to be a consequence of damage to the alveolar epithelium and abnormal wound healing, but it has also been shown that both genetic factors and environmental conditions can significantly contribute to the development of the disease [17,18]. The condition is characterized by subpleural basal fibrosis, honeycomb changes, and collagen and ECM deposition, which ultimately result in life-threatening structural changes in lung tissue and loss of pulmonary ventilation and diffusion function.

Alveolar epithelial damage, caused by external factors (infection, toxins, smoke) or internal factors (inflammation, oxidative stress, abnormal immune response), leads to the release of fibrogenic cytokines, including TGF and tumor necrosis factor (TNF), as well as growth factors such as PDGF and connective tissue growth factor (CTGF) [19]. Elevated levels of these fibrogenic cytokines and growth factors, both locally and systemically, stimulate the activation and proliferation of lung fibroblasts to some extent. Upon activation, fibroblasts differentiate into pulmonary myofibroblasts, which are responsible for the excessive production of ECM proteins in fibrotic lung tissue. These myofibroblasts also regulate the balance between MMP and tissue inhibitors of metalloproteinases (TIMPs), thereby facilitating the process of IPF [20].

### 2.2. The Relationship between Silicosis and IPF

#### 2.2.1. Cause of Disease

The inhalation of free silica dust in the air is the main cause of silicosis. The emergence of silicosis is closely linked to the volume, structure and dimensions of the silica particles inhaled [21], whereas IPF is a particular type of interstitial pneumonia that is fibrosing, chronic and progressive, but for which the origin is yet to be determined [22].

Despite the fact that occupational exposure can induce IPF to some extent, a plethora of epidemiological investigations have demonstrated a significant link between smoking, chronic viral infections, and the genetics of IPF [22,23,24].

#### 2.2.2. Pathogenesis

Extensive research has revealed that the development of silicosis fibrosis is not solely due to one factor, but rather a complex outcome resulting from various factors and links [25]. The primary pathogenic mechanisms of silicosis involve direct cytotoxic effects, the generation of ROS and reactive nitrogen radicals, the release of inflammatory chemokines, the initiation of fibrotic pathways and cell death [4]. The cellular molecule and gene transcription regulation fields are also being explored in relation to silicosis [26]. Similarly, the pathogenesis of IPF has also not been fully elaborated, but relevant studies have shown that oxidative stress, inflammatory response, a series of cytokines and their related signal transduction pathways are also involved in the disease process [27].

Cellular senescence, which includes molecular changes such as telomere shortening, is involved in the pathogenesis of various chronic diseases, including lung diseases [28]. Telomere shortening may be a common causative feature of the development of IPF and silicosis [29,30]. Moreover, epigenetic regulation is a newly discovered mechanism of silicosis and IPF, but further research is necessary to fully understand this process [31,32].

Silicosis and IPF are respiratory diseases that cause damage to the lungs. In response to the inflammatory process, fibroblasts proliferate and produce excessive collagen fibers, leading to the deposition of ECM lung tissue remodeling, ultimately resulting in impaired lung function [33,34]. Moreover, the two diseases share similarities in the upregulation of TGF-β and extracellular signal-regulated kinase (ERK) signaling pathways in cytokine and growth factor pathways, and a relationship with autophagy [35]. While the immediate causes of silicosis and IPF may differ, the overall mechanisms of the subsequent profibrotic reaction are comparable, being characterized by ECM deposition and fibroblast proliferation [36]. Therefore, potential therapeutic drugs for the treatment of silicosis may be sought from IPF.

#### 2.2.3. Symptoms and Complications

Silicosis and IPF share certain symptoms, such as dyspnoea and loss of appetite. However, patients with silicosis may experience additional manifestations, such as chest pain, pulmonary dysfunction, low-grade fever, night sweats, and active shortness of breath, while IPF typically presents as cough and sputum production [37]. In some severe cases, patients with IPF may also exhibit general discomfort, including weakness and joint pain.

The development of silicosis can result in various complications, including tuberculosis, chronic obstructive pneumonia, and rheumatoid arthritis [38]. Similarly, individuals with IPF may experience pulmonary hypertension, acute exacerbation of pulmonary fibrosis, respiratory tract infections, acute coronary syndrome, and thromboembolic disease [39]. However, as the diseases advance, both silicosis and IPF can increase the likelihood of developing lung cancer [40,41,42]. Ultimately, respiratory failure is the primary cause of mortality in patients with these conditions [43]. Consequently, we can investigate the symptomatic treatment drugs used in the treatment of IPF to identify potential therapeutic agents that can alleviate silicosis.

## 3. Treatment of Silicosis and IPF

The clinical management of silicosis remains a challenge, as there is no definitive index for assessing the extent of pulmonary fibrosis and no breakthrough drugs or targeted treatments are available. Whole lung lavage (WLL) has been demonstrated to control the symptoms of silicosis and improve the quality of life, particularly in the early stages of the ailment. Moreover, lung transplant is a well-established treatment for end-stage silicosis [44,45]. Unfortunately, the scarcity of donor lungs, numerous contraindications, and the high risks associated with the surgery all contribute to the difficulty of carrying out transplantation [46]. In recent years, researchers have developed new drugs and new approaches for the treatment of silicosis, but most of the research is still in the basic research stage, including animal experiments and cell experiments.

### 3.1. Drug Research Targeting Mechanisms of Silicosis

The study findings have demonstrated the considerable scope of pharmaceuticals in the treatment of silicosis fibrosis. These drugs can address one or more fundamental mechanisms of silicosis, mitigating the inflammation and/or fibrosis triggered by silica.

#### 3.1.1. Oxidative Stress Response

The oxidative stress encountered during pulmonary fibrosis is closely associated with both the nuclear factor kappa-B (NF-κB) signaling pathway, the kelch-like ECH-associated protein 1 (Keap1)/NF-E2 p45-related factor 2 (Nrf2)/antioxidant-responsive element (ARE) pathway and the NADPH oxidase (NOX) 4-Nrf2 signaling cascade [47]. According to available evidence, it has been suggested that the administration of dioscin, dihydroquercetin, and quercetin may have a beneficial effect on pulmonary fibrosis by impeding the infiltration of macrophages, B lymphocytes, and T lymphocytes into the lung tissue [48,49,50]. Oleanolic acid has been found to be advantageous in the management of pulmonary fibrosis, most likely due to its capacity to diminish serum TNF-α levels, decrease collagen content in lung tissues, and hinder oxidative stress and NF-κB activation [51]. Moreover, the inhibition of nuclear translocation of NF-κB p65 by compounds such as quercetin, bletilla striata polysaccharide, holly and ursolic acid can effectively suppress inflammation and provide some relief from lung fibrosis [51,52,53]. An investigation has uncovered that tanshinone IIA (tan IIA), earthworm extract and emodin possess the ability to effectively suppress the EMT and TGF-β1/Smad signaling pathways. Furthermore, they can reduce silica-induced oxidative stress by activating the Nrf2 signaling pathway [54,55,56,57,58]. Oxidation, and imbalance in oxidation, can be attributed to the fluctuations in the levels of antioxidant enzymes linked to pulmonary fibrosis, as well as ROS catalyzed by NOX2/4. Studies have revealed that white tea extract and astragalus could enhance the level of antioxidant enzymes in lung tissue, strengthen the antioxidant capacity of lung tissue, and exhibit antifibrotic properties [59].

The fibrosis process in silicosis may be impacted by the use of traditional Chinese medicine (TCM) compound preparations that target oxidative stress. A recent study has revealed that the low-dose nebulized inhalation of Chinese herbal preparations can reduce the levels of malondialdehyde (MDA) and interferon γ, while also improving pulmonary fibrosis and inflammation in silicosis [60]. Other compounds, such as panicolin, bletilla striata polysaccharide, small molecule components, and N-acetylcysteine (NAC) have also been found to be effective in controlling the progression of silicosis. These drugs work by modulating the hydroxyproline levels and regulating factors such as superoxide dismutase (SOD) and MDA in the oxidation system [61]. Studies have also provided evidence that hesperetin and panicolin can enhance the activity of antioxidant enzymes including SOD and glutathione peroxidase (GPx) in animals. Furthermore, these compounds can reduce the levels of lipid peroxidation and TNF-α and slow down the progression of pulmonary fibrosis [62]. Moreover, fullerene nanoparticles (FNs) possess an impressive ability to clear ROS, enabling them to effectively inhibit the secretion of mature IL-1β and the influx of neutrophils [63]. It is worth noting that lung administration of FNs does not cause significant toxicity [63]. Research has also shown that combination therapy with astragalus, huangjin and other drugs for pneumoconiosis complicated with chronic obstructive pulmonary disease can effectively reduce SOD, MDA, glutathione peroxidase, and pulmonary fibrosis in patients, thereby impeding the advancement of pulmonary fibrosis [59]. However, in most cases, fibrosis has already begun at the time of diagnosis, so controlling early inflammatory indicators has little effect on reversing the fibrosis.

#### 3.1.2. Autophagy and Apoptosis

Currently, research has shown that apoptosis of AM induced by SiO2 can be regulated by various intracellular pathways, including the mitochondria-mediated intracellular apoptosis program [64], NF-κB signaling pathway [65], factor-related apoptosis (Fas)-mediated exogenous pathway [66], p53 signaling pathway [67], endoplasmic reticulum stress [68], PI3K/AKT/mTOR signaling pathway [69], Janus kinase (JAK)2/signal transducer and activator of transcription (STAT)3 signaling pathway [70], and others. Studies have demonstrated that emodin can inhibit silica-induced apoptosis and exert antifibrotic effects by increasing the expression of anti-apoptotic protein B-cell lymphoma-2 (BCL2) and decreasing the expression of pro-apoptotic protein-BCL2-associated X(Bax) protein [58]. Similarly, dioscin can promote AM autophagy, enhance the clearance of mitochondria damaged by silica dust, and reduce the activation of mitochondria-mediated apoptosis pathway, so that AM resists apoptosis caused by silica dust and reduces the secretion of pro-inflammatory and profibrotic factors [48]. Moreover, AKEX0011, which is based on the phenylpyridone scaffold of pirfenidone, has been found to inhibit the Arabidopsis Serine/Threonine Kinase1 (ASK-1)/P38 signaling pathway and regulate the polarization of macrophages by reducing the phosphorylation of NF-κB/peroxisome proliferator-activated receptor gamma (PPAR-γ) proteins [71]. Atractylenolide III has also been shown to inhibit autophagy through an mTOR-dependent mechanism, improve the blockade of AM autophagy degradation, and reduce silicon-induced AM apoptosis [72]. Additionally, kaempferol has been found to reduce the phosphorylation levels of PI3K, AKT and mTOR proteins in lung tissues of mice with pulmonary fibrosis, thereby inhibiting the PI3K/AKT/mTOR signaling pathway and increasing the level of autophagy to exert antifibrotic effects [73]. Pirfenidone has also been found to reduce pulmonary fibrosis in silicosis rats by reducing IL-17A secretion through the JAK2/STAT3 signaling pathway and inhibiting macrophage polarization [74].

#### 3.1.3. Regulation of Signaling Pathways Related to EMT

TGF-β is a crucial regulator of EMT, which reduces the expression of epithelial markers such as E-cadherin and α-catenin while increasing the expression of mesenchymal markers such as N-cadherin, vimentin, and alpha-smooth muscle actin(α-SMA) [75]. Certain natural compounds such as emodin, quercetin, tadalafil, and sodium ferulate have been found to increase E-cadherin levels and reduce the expression of Vimentin, α-SMA, Col-I, TNF-α, IL-1β, and pro-inflammatory factors TGF-β1, thereby regulating EMT to relieve fibrosis in silicosis [76,77]. Tamoxifen, on the other hand, reduces the serum TGF-β1 content in rats in the model group in a dose-dependent manner, effectively inhibiting the process of silicosis [78], but also exhibits certain hepatotoxic effects. Interestingly, AKEX0011 has also been found to reduce the infiltration of neutrophils and macrophages in lung tissue and decrease the protein levels and mRNA expression of fibrosis-associated proteins [71]. Moreover, natural compounds such as ursolic acid and astragaloside IV can deactivate the mitogen-activated protein kinase (MAPK) signaling pathway, regulate PI3K/AKT/mTOR, inhibit the hyperphosphorylation of forkhead box O3 (FOXO3a) to inhibit EMT, and slow the progression of pulmonary fibrosis [79,80]. Furthermore, bone morphogenetic protein (BMP)-7 can attenuate silica-induced silicosis fibrosis by modulating the balance between TGF-β/Smad and BMP-7/Smad signaling pathways, but its preventive effect is more significant than its therapeutic effect [81]. It is worth noting that TGF-β1-mediated Smad-dependent pathways and Smad-independent pathways can independently regulate the fibrotic response or exert their effects through interaction with Smad proteins. Buyang Huanwu Tang has been found to regulate multiple signaling pathways, including TGF-β1/Smad2/3 and PIK3/AKT, resulting in a reduction of pulmonary fibrosis and inflammatory changes in silicosis [82]. Similarly, Dahuang Zhechong pills have been shown to effectively inhibit the p38 MAPK/NF-κB/TGF-β1 pathway and TGF-β1/Smad pathway, thereby reducing and eliminating the persistent inflammatory stimulation caused by SiO2 in the lungs [83]. Natural medicines work through these pathways for different periods and have different antifibrotic effects. However, ensuring that a single drug or combination of drugs works in the body without damaging other functions remains a complex problem that requires further investigation.

#### 3.1.4. Blocking Silicosis Fibrosis by Targeting Fibroblasts

Pulmonary fibrosis is characterized by excessive secretion of collagen, fibronectin, and elastin by myofibroblasts, leading to an imbalance in MMP/TIMP regulation and accumulation of the ECM [84]. Quercetin has been found to reduce the expression of MMP-2 and MMP-9, thereby inhibiting the formation of the ECM and reducing pulmonary fibrosis [85,86,87]. Schisandrin B has also been shown to inhibit the expression of MMP-2, slowing the onset of pulmonary fibrosis in rats induced by silica [88]. Additionally, gallus domesticus extract has been found to reduce the content of FNs, Col-I, MMP-9 and MMP-12 in bronchoalveolar lavage fluid (BALF) in rats [89]. Dasatinib has been found to induce macrophage bias towards the M2 macrophages phenotype, improving lung mechanics in mouse models of acute silicosis by downregulating the expression of IL-1β, TNF-α and TGF-β proteins in lung tissue and upregulating the expression of arginase and MMP-9 [90]. While drugs can regulate fibroblast activation and ECM degradation through multiple signal transduction pathways, they cannot reverse transformed myofibroblasts and cure silicosis.

#### 3.1.5. Other Mechanisms to Prevent and Treat Silicosis Fibrosis

In addition to the aforementioned studies, researchers have been investigating alternative mechanisms through which drugs can affect silicosis fibrosis. For instance, kaempferol has been found to restore silica-induced microtubule-associated protein 1A/1B-light chain 3 (LC3) lipidation without increasing p62 levels [73]. Similarly, metformin has been shown to inhibit the inflammatory response in macrophages, reduce TGF-β1-stimulated fibroblast activation in lung fibroblasts via the AMPK-dependent pathway, increase the expression levels of p-AMPK, LC3B and Beclin1 proteins, reduce the levels of phosphorylated mammalian rapamycin (p-mTOR) and p62 proteins in vivo, and activate autophagy to inhibit silica-induced pulmonary fibrosis [91]. Quercetin reduces the accumulation of myofibroblasts and restores fibroblast apoptosis sensitivity by upregulating the Fas ligand receptor and the expression of caveolin-1 [52]. Dihydroquercetin inhibits ferritin phagocytosis-mediated iron death in cells and improves silica-induced pulmonary fibrosis [49], while lysophosphatidylcholine acyltransferase 1 (LPCAT1) alters the balance between phosphatidylcholine and lysophosphatidylcholine and inhibits the development of silicosis in mice [92].

Moreover, the combination of tetrandrine tablets and matrine injection has been shown to have a low incidence of adverse reactions and to be able to improve lung ventilation function, alleviate symptoms, and exhibit significant clinical value [93]. The combined use of desipramine and NAC has been found to effectively suppress the inflammatory response and delay the progression of silicosis fibrosis in a synergistic manner [61]. This synergy has also been observed with the combination of NAC and tetrandrine [94]. However, drug combinations not only show synergistic effects, but also are accompanied by the side effects of multiple drugs.

In summary, it has been discovered that numerous drug components and TCM compound preparations can hinder the development of silicosis fibrosis by intervening in the silicosis TGF-β/Smad signaling pathway, oxidative stress mechanism, apoptosis, and autophagy. Additionally, various pathways have been observed to interact with each other, such as emodin, tan IIA, curcumin, and other drugs, which can exert anti-inflammatory and antifibrotic effects through different intervention mechanisms [95]. In the future, apart from conducting more in-depth research on the pathogenesis and progression mechanism of silicosis, it is vital to focus on the interaction between signaling pathways and drug intervention targets, and to prioritize the transformation of basic research into clinical practice. Furthermore, it is essential to conduct further research on the diagnosis and treatment technology of TCM. A specific and standardized diagnosis and treatment plan for the combination of traditional and modern medicine for the treatment of silicosis should be established as soon as possible. Individualized treatment plans should be formulated based on the patients’ specific conditions to improve their quality of life, reduce economic pressure, and promote their return to social life faster and more effectively.

### 3.2. Antifibrosis Treatment Drugs for IPF

Numerous studies have been found that by inhibiting the NF-κB signaling pathway, parthenolide, hesperidin and dehydrocostus can inhibit the early inflammatory response, thereby exerting an antifibrotic effect [96,97,98]. Moreover, sulforaphane, dihydroartemisinin, melatonin and ginkgo biloba extract have been found to reduce oxidative stress to exert anti-pulmonary-fibrosis effects [99,100,101,102,103]. Similarly, the total extract of Yupingfeng and the combination of tan IIA and puerarin can exert antifibrotic effects by regulating the PI3K/AKT/mTOR signaling pathway and JAK/STAT signaling pathway [104,105,106]. As for regulating signaling pathways related to EMT, asiatic acid and oridonin inhibit the expression of TGF-β1 in lung tissue, accompanied by a decrease in Col-I, Col-III, α-SMA and TIMP-1, as well as inactivation of Smads and ERK1/2 [107,108]. Moreover, juglanin and Rhodiola rosea L. significantly attenuate bleomycin (BLM)-induced pulmonary fibrosis by reducing the expression of fibrotic features such as TGF-β1, fibronectin, MMP-9, α-SMA, and Col-I [109], particularly in a dose-dependent manner in the case of Rhodiola rosea L [110]. Further investigation indicated that tannic acid treatment suppresses BLM-induced phosphorylation of ERK1/2 in lungs [111]. Moreover, nimbolide regulates autophagy signaling by inhibiting LC3 and p62 expression and increasing Beclin-1 expression [112]. Studies have found that treatment with Olanpingensis polysaccharides can alleviate pulmonary fibrosis progression, mainly by reducing the recruitment of macrophages to the lungs [113].

Numerous studies have demonstrated the potential of various TCMs, including Xin Jia Xuan Bai Cheng Qi Decoction extract, Renshen Pingfei Decoction, Yangyin Yiqi Mixture, and Qing-Xuan Granules, to ameliorate the progression of pulmonary fibrosis. The mechanism may involve the suppression of the TGF-β1/Smad signal pathway and EMT in BLM-induced pulmonary fibrosis [114,115,116,117]. Other drugs, such as Feifukang, Modified Kushen Gancao Formula, Jinshui Huanxian Formula, and Yifei Sanjie Formula, target different pathways to alleviate pulmonary fibrosis [118,119,120,121]. Combination therapies involving traditional Chinese and modern medicines have also been studied. For example, Salvia Miltiorrhiza and Ligustrazine have been found to be safe and effective in repressing BLM-induced pulmonary fibrosis, possibly by modulating the expression of TNF-α and TGF-β1 [122]. Dexamethasone combined with berberine has also shown antifibrotic effects, which involve inhibiting C-X-C motif chemokine ligand (CXCL)-14 and MMP-2/MMP-9 expression and preventing the activation of Smad2/3 and hedgehog signaling pathways [123].

Since there are similarities in the mechanisms of silicosis and IPF, and some drugs have been shown to have a role in both silicosis and IPF animal trials (Table 1), it is worth considering the option of obtaining treatment drugs for silicosis from those used for IPF.

### 3.3. Other Potential Therapies

#### 3.3.1. Stem Cell Therapy

Mesenchymal stem cells (MSCs) are considered relatively safe due to their abundant sources, ease of isolation and culture, low immunogenicity, and secretion factors that can reduce inflammation [130]. Therefore, stem cell therapy has shown significant promise in the treatment of silicosis. Preclinical studies have shown that the administration of MSCs through endotracheal or tail vein effectively inhibits inflammation and fibrosis in mouse models of silicosis, leading to therapeutic benefits [131,132]. Moreover, previous research suggests that N-acetyl-seryl-aspartyl-lysyl-proline (Ac-SDKP) may alleviate the fibrotic symptoms of silicosis by regulating endoplasmic reticulum stress [133]. Averyanov’s study demonstrated that transplantation of bone-marrow-derived MSCs at high cumulative doses (1.6 × 10^9^/mL) is well tolerated and safe in patients with pulmonary fibrosis, with only minor side effects such as fever. These findings confirm the safety, tolerability, and potential benefits of high-dose MSCs, and may pave the way for future stem cell transplantation trials [134]. In recent years, induced pluripotent stem cells have gained popularity in the research community. These cells can be used to cultivate numerous mature cell types and create lung organoids that can aid in the development of personalized treatments for silicosis [135].

Although this approach has potential, there are several issues that need to be resolved. Firstly, ongoing assessment of the clinical safety and effectiveness of these pharmacotherapies is necessary. Secondly, stem cell therapy is still in its early stages and faces numerous technical challenges. Thirdly, the process of conducting clinical trials and addressing ethical considerations for new drug targets can be lengthy. Lastly, the exorbitant cost of personalized and experimental therapies limits their widespread clinical application due to financial constraints.

#### 3.3.2. Antifibrotic Target Therapy

The advent of high-throughput omics technology and its amalgamation with bioinformatics has brought about a paradigm shift in gene editing technology, presenting a viable remedy for investigating the causation and management of silicosis.

Recent genomic analyses have revealed that certain single nucleotide polymorphisms (SNPs) are connected to an increased risk of silicosis, particularly rs73329476 and rs12812500, which are linked to the development of pneumoconiosis [136]. By conducting weighted gene co-expression network analysis, Jiaqi Lv et al. identified silicosis-related modules and pivot genes and found that the Hippo signaling pathway plays a beneficial role in silicosis fibrosis. This discovery has helped to elucidate the precise mechanisms of silica-induced pulmonary fibrosis and identify the molecular initiation events and adverse outcome pathways of silicosis [137]. In the field of epigenomics, research has demonstrated that certain miRNAs are significantly linked to the development and progression of pneumoconiosis and may serve as non-invasive biomarkers and prognostic indicators for early pneumoconiosis. For example, miRNA-29, which regulates the Wnt/β-catenin pathway, and miRNA-326, which targets tumor necrosis factor superfamily-14 and polypyrimidine bundle-binding proteins, have been shown to inhibit fibrosis in silicosis [138,139]. Furthermore, miRNAs have serum-detectable, morphologically stable, and reusable freeze-thaw properties, making them promising biomarkers for the early diagnosis of silicosis [140]. Finally, Lv-shβ-catenin has been found to reduce the expression of MMP-2 and MMP-9, reducing pulmonary fibrosis and inhibiting the formation of ECM [85,87]. These findings have important implications for the prevention and treatment of silicosis and highlight the potential of genomic and epigenomic research to improve our understanding of this debilitating disease.

The use of transcriptomics and proteomics has led to the identification of a variety of protein-coding genes and proteins that are differentially expressed in silicosis [141]. These findings suggest potential targets and signaling pathways that may play a significant role in lung disease. Bo C and Mingyao Wang et al. conducted a study using different clusters for pathway enrichment. They classified differentially significant proteins and found that although differential expression features in the omics datasets are involved in different pathways, these features are found in some key signaling pathways, such as inflammatory response and interstitial fibrosis regulation [26,142]. Moreover, according to the findings of a functional analysis conducted by Yingdie Zhang et al., 18 m6A-mediated mRNAs were observed to regulate pathways that were closely related to “phagosomes”, “antigen processing and presentation”, and “apoptosis”. This suggests that m6A methylation plays an essential role in the development of silicosis [143]. Multiomics analysis can help identify potential drug targets for the treatment of silicosis fibrosis, and this approach can be used to design and develop targeted drugs. However, these targets are still in the preclinical stage and require further clinical experimental studies before they can be applied in the treatment of silicosis.

In the future, the utilization of novel omics techniques in the study of silicosis, including the integration of omics and spatomics, may establish a more comprehensive and objective approach, significantly enhancing our capacity to explore the pathology and molecular mechanisms of silicosis and offering a prospect for discovering effective diagnostic and therapeutic interventions for silicosis. Nonetheless, it is imperative to subject the targets identified by this method to rigorous testing and validation before implementing them in clinical practice. It is essential to note that the pathological mechanisms of silicosis are intricate, and the biological effects of cytokines produced by inflammatory cells interact in a complex manner, leading to the formation of a network of cells. Consequently, blocking a single target may not suffice to alter the degree of inflammation and fibrosis and effectively treat the disease. Thus, a comprehensive understanding of the pathological mechanisms of silicosis is indispensable.

### 3.4. Clinical Drugs of Silicosis and IPF

To date, there is no specific remedy for silicosis. The efficacious treatments available for patients with silicosis are WLL and lung transplantation. We should not only attach importance to basic research, but also accelerate the process of clinical trials of basic research results.

#### 3.4.1. Silicosis

Management of silicosis consists of using bronchodilators and cough medication. However, symptomatic treatment may only ameliorate symptoms rather than restoring health. Most drugs that have shown positive effects in animal models, especially reducing lung fibrosis, have not been yet translated into clinically approved drugs in many countries, including Europe and the USA [144]. Tetrandrine stands as the sole drug sanctioned for silicosis treatment in China, as per the approval of regulatory authorities [145]. Recent clinical trials have demonstrated that the combined administration of tetrandrine with other drugs yields more pronounced therapeutic outcomes than conventional treatment modalities [94]. Despite the minimal adverse effects of tetrandrine therapy for pulmonary fibrosis, it remains incapable of reversing fibrosis and curing patients afflicted with silicosis.

#### 3.4.2. IPF

Since many of the signaling pathways of silicosis and IPF overlap, it is possible that clinical medications used in the treatment of IPF could hold promise in the treatment of silicosis. For instance, antifibrotic drugs like pirfenidone and nintedanib, which are commonly prescribed for IPF [146], have demonstrated efficacy in reducing lung inflammation and fibrotic changes in animal models of silicosis [147,148,149]. Due to the significant side effects of nintedanib, it has not been an option for silicosis treatment, but recently an experiment engineered a nanocrystal-based suspension formulation of nintedanib possessing specific physicochemical properties to enhance drug retention in the lung for localized treatment of silicosis [150], bringing new hope to nintedanib for treating silicosis and highlighting that overcoming the side effects of existing clinical drugs can provide a new direction for the treatment of silicosis. Therefore, insights gained from antifibrosis drugs utilized in IPF could offer potential avenues for treating silicosis.

While these drug studies may only serve to delay the progression or alleviate the clinical symptoms of silicosis, they nevertheless provide a glimmer of hope for a potential cure. The treatment of silicosis is a protracted process that necessitates persistent basic research, comprehensive exploration of pathogenesis, implementation of advanced biological science technology, and other related approaches. In this manner, we can discover more drugs that can be utilized in clinical practice, thereby increasing the likelihood of curing silicosis.

## 4. Conclusions

The treatment of silicosis fibrosis has been a challenging topic due to the differences between animal models and human bodies and other difficulties, and there is currently no effective treatment available. In recent years, scholars have made some progress in the treatment of silicosis pulmonary fibrosis, which has brought hope for further exploration of the therapeutic effect of drugs and the clinical treatment of silicosis. However, the mechanism of action and side effects of some drugs for silicosis pulmonary fibrosis remain unclear, which limits their clinical application. The direction of future research may involve delving into innovative approaches to tackling silicosis through the implementation of combination administration or multitarget therapy, supplemented by state-of-the-art biological science technology. Additionally, drawing inspiration from comparable diseases like IPF may prove to be a valuable avenue of exploration. This approach can help to bridge the gap between basic research and clinical application and shorten the distance between research and practice for silicosis.

## Figures and Tables

**Figure 1 ijms-24-08333-f001:**
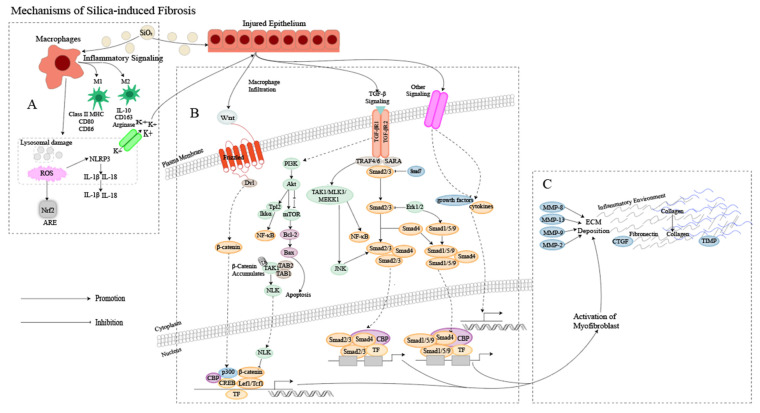
Mechanism of silica-induced fibrosis. (**A**) Alveolar macrophages (AMs) engulf silica dust, causing them to turn into dust cells. Subsequently, AMs may synergize with alveolar epithelial cells (AECs) to release a large amount of ROS to participate in oxidative stress reactions, activate NOD-like receptor thermal protein domain associated protein 3 (NLRP3) inflammatory bodies through lysosomal damage and potassium outflux, and activate the release of inflammatory mediator interleukin (IL) -1β, IL-18 and other cytokines inducing epithelial–mesenchymal transition (EMT). Meanwhile, AMs can polarize into M1 and M2 types, playing a role in promoting inflammation, fibrosis, and antigen presentation, increasing the proliferation of lung fibroblasts and collagen synthesis and secretion, and promoting the formation of fibrosis through apoptosis and autophagy. (**B**) Ongoing damage and damage to lung cells by silica lead to pathological overdeposition of extracellular matrix (ECM) proteins accompanied by upregulation of myofibroblast activity, resulting in a chronic inflammatory environment of macrophage and immune cell infiltration. In this cellular environment, cytokines and growth factors are released in large quantities, activating many signaling cascades, including members of the transforming growth factor-beta (TGF-β) family and Wingless/Int (Wnt) 1, the phosphatidylinositol 3-kinase (PI3K)/protein kinase B (AKT)/mammalian target of rapamycin (mTOR) pathway and other pathways. (**C**) Fibroblasts then aggregate in the area of injury, and the combination of ECM degradation by MMPs and excessive collagen deposition leads to granuloma formation and lung tissue remodeling.

**Table 1 ijms-24-08333-t001:** Drug candidates for the treatment of silicosis.

Name	Structure	Source	Therapeutic Target	Experiment Model	Silicosis	IPF	Ref.
Dioscin	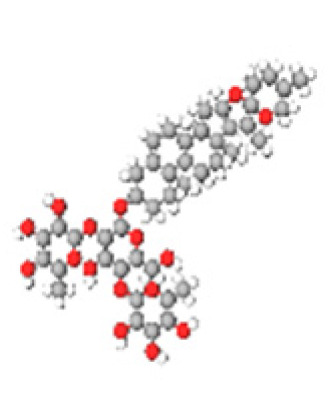	Dioscorea nipponica Makin	Reducing pro-inflammation and pro-fibrotic cytokine secretion and inhibiting TGF-β/Smad signaling and fibroblast activation.	In vivo:Silica- induced PF mice BLM-induced IPF miceIn vitro:RAW264.7 and NIH-3T3 cell	+	+	[48,124]
Dihydroquercetin	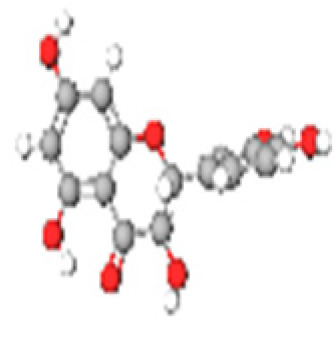	Yew, larch and cedrus brevifolia bark	Inhibiting ferroptosis signaling pathway and modulating FOXO3-mediated NF-κB signaling to attenuate pulmonary fibrosis.	In vivo:Silica- induced PF miceIn vitro:human bronchial epithelial cells	+	−	[49]
Quercetin	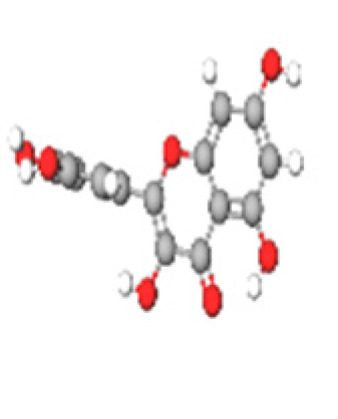	Diverse plantsources	Decreasing expressions of the senescence-associated secretory phenotype, including proinflammatory factors, Il-6, TNF-α, TGF-β and MMP, and modulating the redox balance by inducing Nrf2.	In vivo:Silica- induced PF mice and ratsIn vitro:RAW264.7 macrophages with silicaEx vivo:blood of IPF patients	+	+	[50,125]
Oleanolic acid	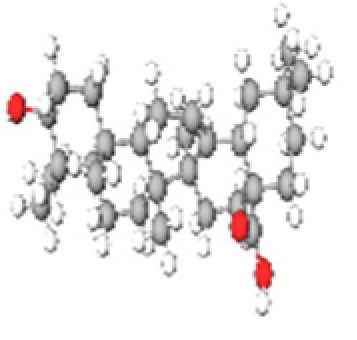	Vegetables andfruits	Modulating the AKT/NF-κB pathway to decrease the expression of cytokines and collagen.	In vivo:Silica- induced PF rats	+	−	[51]
NAC	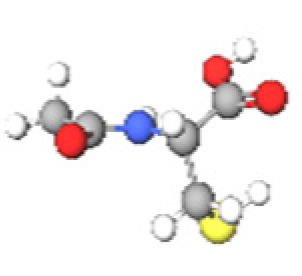	Allium plants	Alleviating the inflammatory response through non-PI3K/AKT/mTOR signaling pathway.	In vivo:Silica- induced PF mice and ratsPatients with silicosis and IPF	+	+	[61]
Tan IIA	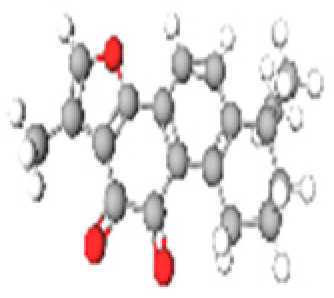	Salviamiltiorrhiza	Inhibiting the EMT and TGF-β1/Smad signaling pathway, reducing oxidative stress, activating the Keap1/Nrf2/ARE signaling pathway.	In vivo:Silica- induced PF ratsBLM-induced IPF mice and rats	+	+	[54,55,56,106]
Earthworm extract	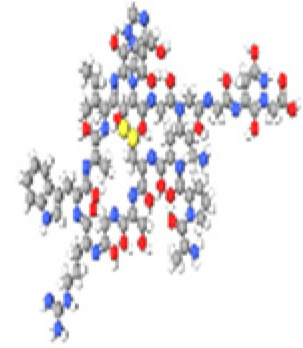	Earthworm	Inhibiting oxidative stress, mitochondrial apoptotic pathway and EMT.	In vivo:Silica- induced PF mice	+	−	[57]
Emodin	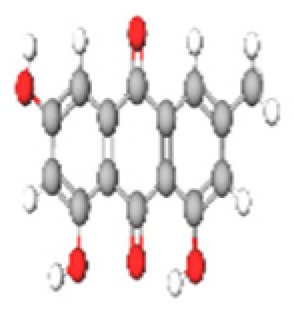	Rhubarb	Inhibiting EMT, TGF-β1/Smad2/3 and NF-κB pathway to attenuate pulmonary fibrosis.	In vivo:Silica- induced PF miceBLM-induced IPF ratsIn vitro:human macrophages and AECs	+	+	[58,126]
Kaempferol	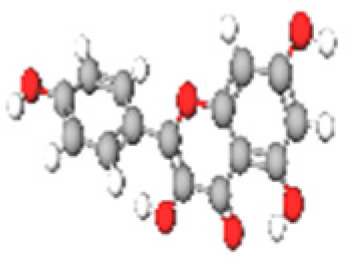	plants and fruits	Restoring the lipidation of LC3 without affecting p62 expression, increasing autophagic flux, ameliorating silica-induced pulmonary fibrosis.	In vivo:Silica- induced PF mice	+	−	[73]
Tadalafil	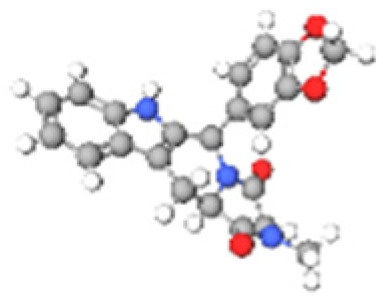	Phosphodiesterase type V (PDE 5) inhibitors	Down-regulating inflammatory and fibrogenic cytokines expression, restorating oxidants/antioxidant hemostasis, antioxidant boost and promoting of angiogenesis	In vivo:Silica- induced PF rats	+	−	[76]
Sodium ferulate	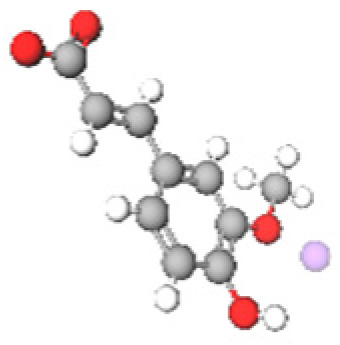	Ferulic acid	Inhibiting lung injury and fibrosis through the neutrophil alkaline phosphatase 3(NALP3)/TGF-β1/α-SMA pathway.	In vivo:Silica- induced PF mice	+	−	[77]
Tamoxifen	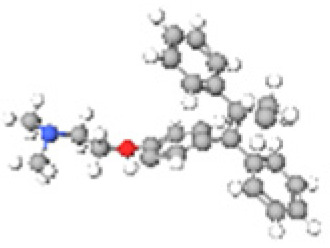	Selective estrogen receptor modulator	Decreasing lung fibrosis score with blood TGF-β levels.	In vivo:Silica- induced PF rats	+	−	[78]
Atractylenolide III	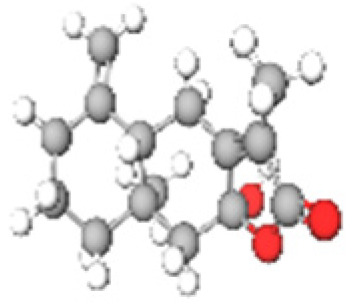	Leucodon rhizome	Inhibiting autophagy by mTOR-dependent manner and improving the blockage of autophagic degradation in AMs.	AMs in human silicosis	+	−	[72]
Schisandrin B	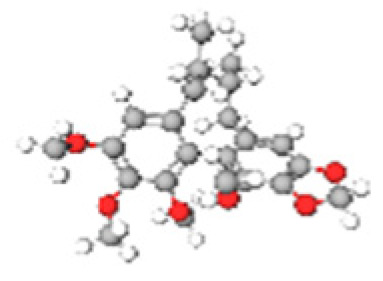	Schisandrachinensis	Reducing the pulmonary fibrosis through inhibition of the mRNA express of TGF-β1 and Smad4 in the lung tissue, modulating the TGF-β1/Smad4 signal transduction pathway and inhibiting the target gene activation.	In vivo:Silica- induced PF rats	+	−	[88]
Dasatinib	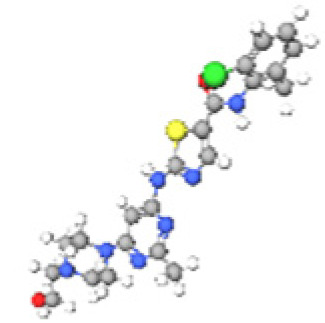	Tyrosine kinase inhibitor	Inducing macrophage polarization toward the M2 phenotype and reducing lung inflammation and fibrosis	In vivo:Silica- induced PF mice	+	−	[90]
Metformin	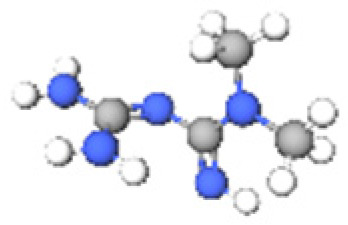	French Lilac	Alleviating inflammatory response and collagen deposition in the process of pulmonary fibrosis via suppressing EndoMT through the AMPK/mTOR signaling pathway.	In vivo:Silica- induced PF mice and rats	+	−	[91]
Curcumin	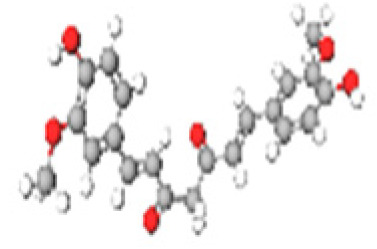	Turmeric	Inhibiting the expression of cyclooxygenase (COX) 2, NF-κB p65, fibronectin and p-AMPK and IL-17A induced inflammation and pulmonary fibrosis through the AMPKα/COX-2 signaling pathway	In vivo:Silica- induced PF miceBLM-induced IPF mice	+	+	[95,127]
Hesperetin	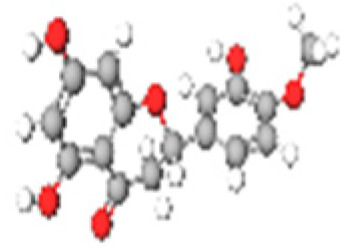	Citrus fruits	Reducing oxidative damage and the inflammatory response to attenuate lung injury.	In vivo:Silica- induced PF rats	+	−	[62]
Oridonin	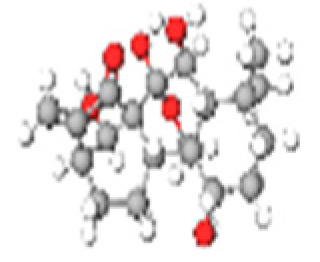	Rabdosia rubescens	Attenuating lung inflammation and fibrosis via covalent targeting inducible nitric oxide synthase (iNOS) and by regulating TGF-β/Smad pathway.	In vivo:silica-induced PF miceBLM-induced IPF mice	+	+	[108,128]
Astragalus polysaccharides	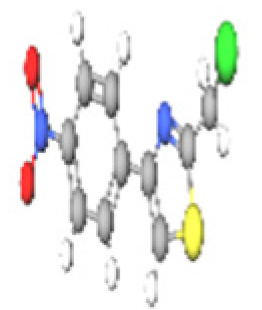	Astragalus	Inhibiting EMT, ROS, TGF-β1/Smads, apoptosis, inflammation pathways.	In vivo:BLM-induced IPF mice and rats	−	+	[59]
mangiferin	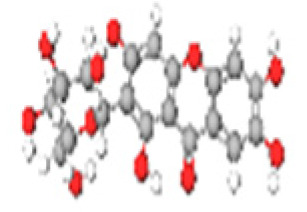	Mango and papaya	Inhibiting toll-like receptor 4 (TLR4)/p65 and TGF-β1/smad 2/3 pathway and reducing NF-κB to attenuate pulmonary fibrosis.	In vivo:BLM-induced IPF mice	−	+	[129]
Parthenolide	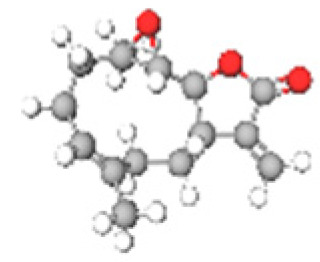	Chrysanthemum parthenium L	Attenuating pulmonary fibrosis via the NF-κB/Snail signaling pathway	In vivo:BLM-induced IPF mice	−	+	[96]
Hesperidin	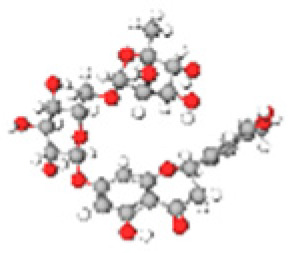	Citrus fruits	Ameliorating pulmonary fibrosis via inhibition of TGF-β1/Smad3/AMPK and IκBα/NF-κB pathways.	In vivo:BLM-induced IPF rats	−	+	[97]
Dehydrocostus Lactone	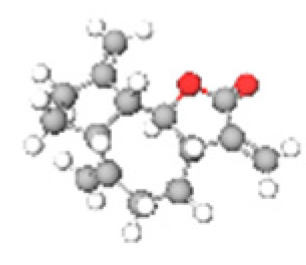	Vladimiria souliei	Inhibiting pulmonary fibrosis and inflammation in mice via the c-Jun NH(2)-terminal kinases(JNK) and p38 MAPK-mediated NF-κB signaling pathways.	In vivo:BLM-induced IPF mice	−	+	[98]
Sulforaphane	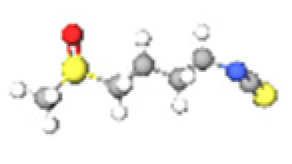	Myrosinase enzyme	Enriching transcriptome of lung mitochondrial energy metabolism and providing pulmonary injury protection via Nrf2.	In vivo:BLM-induced IPF mice	−	+	[102]
Dihydroartemisinin	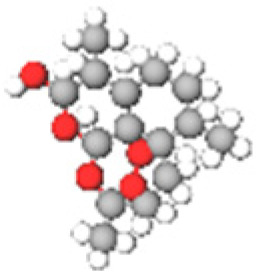	Artemisinin	Suppressing NF-κB signaling in an Nrf2-dependent manner to alleviate oxidative stress.	In vivo:BLM-induced IPF rats	−	+	[103]
Melatonin	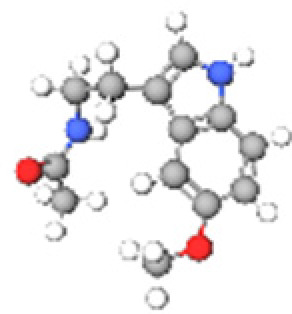	Animals, plants, fungi and bacteria	Ameliorating pulmonary fibrosis via activating Nrf2 and inhibited galectin-3 expression.	In vivo:BLM-induced IPF miceIn vitro:human fetal lung fibroblast1 (HFL1) cells	−	+	[100]
Ginkgo biloba Extract (EGb761)	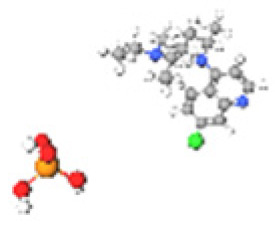	Ginkgo Biloba	Attenuating pulmonary fibrosis by regulating the balance of M1/M2 macrophages and NF-κB-mediated cellular apoptosis.	In vivo:BLM-induced IPF mice and rats	−	+	[101]
Asiatic acid	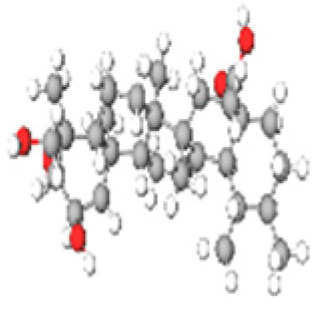	Centella asiatica	Ameliorating pulmonary fibrosis via suppressing pro-fibrotic and inflammatory signaling pathways.	In vivo:BLM-induced IPF mice	−	+	[107]
Juglanin	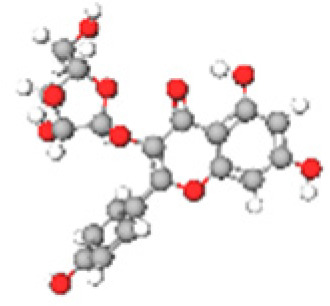	Natural flavonoids	Alleviating lung injury by suppressing inflammation and fibrosis via targeting sting signaling.	In vivo:BLM-induced IPF mice	−	+	[109]
Tannic acid	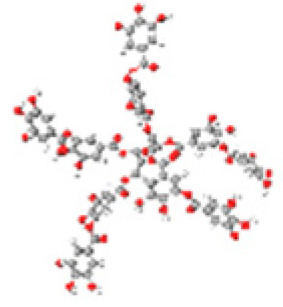	Soft fruits, nuts and other plant tissues	Attenuating TGF-β1-induced EMT by effectively intervening TGF-β signaling.	In vivo:BLM-induced IPF mice	−	+	[111]
Nimbolide	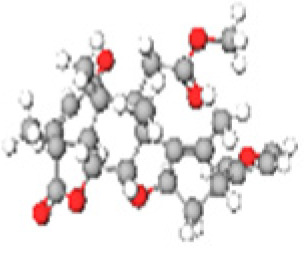	Azadirachta indica L	Ameliorating pulmonary fibrosis through attenuation of TGF-β1 driven EMT.	In vivo:BLM-induced IPF mice	−	+	[112]
Olanpingensis polysaccharides	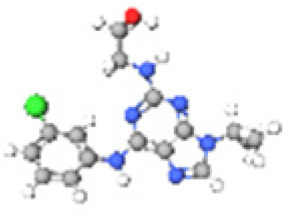	Entomogenous fungi	Alleviating pulmonary fibrosis progression through reducing the recruitment of macrophages to the lungs.	In vivo:BLM-induced IPF mice	−	+	[113]
Berberine	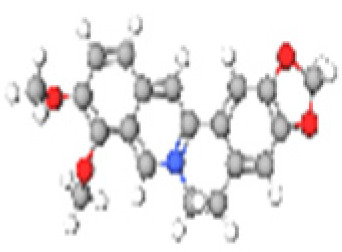	Coptis	Inhibiting Smad and non-Smad signaling cascades and enhanced autophagy against pulmonary fibrosis.	In vivo:BLM-induced IPF rats	−	+	[123]

“−” indicates that no relevant experiments have been done; “+” indicates that the relevant experiment have been done. The molecular structures of active compounds were generated from the online repository www.molview.org, accessed on 15 March 2023. Here, colors in the molecules denote each atom: gray = carbon, white = hydrogen, red = oxygen, blue = nitrogen, and yellow = sulfur.

## Data Availability

Not applicable.

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
