# Peer review of "From Basic Research to Clinical Practice: Considerations for Treatment Drugs for Silicosis"

_ijms, 2023, doi:10.3390/ijms24098333_

Round 1
Reviewer 1 Report
Li et al wrote a review in which the authors compared the relationship between silicosis and IPF and discusses their pathogenesis. The authors also provided a list of drugs that has been research in the context of silicosis and IPF.
The review is novel and comprehensive, bit is a bit dense from time to time due to the amount pathways discusses.
I have a few minor comments that should be address.
Please clarify line 70 about the synergy between AM and AECs. might miss the word "might".
Figure 1 has a lot of information, maybe would be wise to split it up in smaller images and add to the relevant parts of the manuscript when pathways are discussed.
line 113, "around the IPF" not sure what the authors mean with this.
line 138, ralease instead of release.
In addition, i do miss senescence in the manuscript. it is such a big part of the pathology in IPF and it would be great to see what is known about senescence in silicosis.
Quality of english language is good.
Author Response
Dear Reviewer:
Thank you for your letter and reviewer’s comments concerning our manuscript entitled “From basic research to clinical: Considerations for treatment drugs for silicosis” (ID: ijms-2361503). These comments are all valuable and very helpful for revising and improving our paper, as well as the important guiding significance to our review. We have studied comments carefully and have made a correction which we hope meet with approval. Revised portions are marked in yellow on the paper. The main corrections in the paper and the responses to the reviewer’s comments are as follows:
Point 1. Please clarify line 70 about the synergy between AM and AECs. might miss the word "might".
Response 1:
We’re very sorry for the mistake. We have revised this sentence again and marked it in yellow.
Revised:
Subsequently, AMs might synergize with alveolar epithelial cells (AECs) to release a large amount of ROS to participate in oxidative stress reactions, activate NOD-like receptor thermal protein domain associated protein 3 (NLRP3) inflammatory bodies through lysosomal damage and potassium outflux, and activate the release of inflammatory mediator interleukin (IL) -1β, IL-18 and other cytokines inducing epithelial-mesenchymal transition (EMT).
Point 2. Figure 1 has a lot of information, maybe would be wise to split it up in smaller images and add to the relevant parts of the manuscript when pathways are discussed.
Response 2:
Thanks for your sincere suggestion. We have been split the figure in three parts in order to be able to better relate to the following content when pathways are discussed.
Revised:
Please refer to the attached document
Point 3. Line 113, "around the IPF" not sure what the authors mean with this.
Response 3:
We’re very sorry for this "around the IPF" which may have caused confusion. Our original intention was to emphasize the pathological mechanism of IPF, so we have removed "around the IPF" to better clarify this sentence and marked them in yellow.
Revised:
The condition is characterized by subpleural, basal fibrosis, honeycomb changes, collagen and ECM deposition, which ultimately results in life-threatening structural changes in lung tissue and loss of pulmonary ventilation and diffusion function.
Point 4. Line 138, ralease instead of release.
Response 4:
We’re very sorry for the mistake in this word. We have made modifications and marked them in yellow.
Revised:
The primary pathogenic mechanisms of silicosis involve direct cytotoxic effects, the generation of ROS and reactive nitrogen radicals, the release of inflammatory and chemokines, the initiation of fibrotic pathways and cell death.
Point 5. In addition, i do miss senescence in the manuscript. it is such a big part of the pathology in IPF and it would be great to see what is known about senescence in silicosis.
Response 5:
Thanks for your sincere suggestion. Senescence is a multifactorial process that includes molecular changes such as telomere shortening. In this paper, we mentioned a little about telomere shortening in relation to IPF and silicosis, but did not highlight the association between senescence and the two diseases. There are mainly the following reasons: first, the focus of this review is mainly on the analysis of therapeutic drugs, so there is not much emphasis on the analysis of the relationship between the two diseases; Second, because we have studied relatively little in the field of senescence, we dare not discuss this aspect at a deeper level. However, in future research, we will pay more attention to the relationship between senescence and silicosis. In addition, in our article, we have also made appropriate additions to the relationship between senescence and two diseases, and marked them in yellow.
Revised:
Cellular senescence, which includes molecular changes such as telomere shortening, is involved in the pathogenesis of various chronic diseases, including lung diseases [28].

Reviewer 2 Report
Figure 1 is too confusing- it is a simplified schematic anyway and should convey the main points that the authors want to make.
The anti-fibrosis treatment of IPF section also has a paragraph on treatment of silicosis. The silicosis paragraph needs to be separate to improve clarity.
Section 3.4 is called clinical drugs of silicosis and IPF. Maybe the authors mean approved drugs in China for silicosis and IPF. Because this is a review article for a global audience, it would make sense to broaden the scope and include drugs approved in Europe / USA
Author Response
Dear Reviewer:
Thank you for your letter and reviewer’s comments concerning our manuscript entitled “From basic research to clinical: Considerations for treatment drugs for silicosis” (ID: ijms-2361503). These comments are all valuable and very helpful for revising and improving our paper, as well as the important guiding significance to our review. We have studied comments carefully and have made a correction which we hope meet with approval. Revised portions are marked in yellow on the paper. The main corrections in the paper and the responses to the reviewer’s comments are as follows:
Point 1. Figure 1 is too confusing- it is a simplified schematic anyway and should convey the main points that the authors want to make.
Response 1:
Thanks for your sincere suggestion. We have split the figure into three parts to better relate them to the following content when discussing pathways. We have then reinterpreted this figure and marked them in yellow.
Revised:
Please refer to the attached document.
Figure 1. Mechanism of silica-induced fibrosis. A: Alveolar macrophages (AMs) engulf silica dust, causing them to turn into dust cells. Subsequently, AMs might synergize with alveolar epithelial cells (AECs) to release a large amount of ROS to participate in oxidative stress reactions, activate NOD-like receptor thermal protein domain associated protein 3 (NLRP3) inflammatory bodies through lysosomal damage and potassium outflux, and activate the release of inflammatory me-diator interleukin (IL) -1β, IL-18 and other cytokines inducing epithelial-mesenchymal transition (EMT). Meanwhile, AMs can polarize into M1 and M2 types, playing a role in promoting in-flammation, fibrosis, and antigen presentation, increasing the proliferation of lung fibroblasts and collagen synthesis and secretion, and promote the formation of fibrosis through apoptosis and autophagy. B: Ongoing damage and damage to lung cells by silica lead to pathological over-deposition of extracellular matrix (ECM) proteins, accompanied by upregulation of myofibroblast activity, resulting in a chronic inflammatory environment of macrophage and immune cell infiltration. In this cellular environment, cytokines and growth factors are released in large quantities, activating many signaling cascades, including members of the transforming growth factor-beta (TGF-β) family and Wingless/Int (Wnt) 1, the phosphatidylinositol 3-kinase (PI3K) / protein kinase B (AKT)/ mammalian target of rapamycin (mTOR) pathway and other pathways. C: Fibroblasts then aggregate in the area of injury, and the combination of ECM degradation by MMPs and excessive collagen deposition leads to granuloma formation and lung tissue remodeling.
Point 2. The anti-fibrosis treatment of IPF section also has a paragraph on treatment of silicosis. The silicosis paragraph needs to be separate to improve clarity.
Response 2:
Thanks for your sincere suggestion. We realized that there was too much focus on silicosis in the anti-fibrosis treatment of IPF section. Therefore, we have removed this part and only discussed necessary information about silicosis and IPF drug regimens, and then marked them in yellow.
Revised:
Dexamethasone combined with berberine has also been shown antifibrotic effects, which involve inhibiting C-X-C motif chemokine ligand (CXCL)-14 and MMP-2/MMP-9 expression and preventing the activation of Smad2/3 and hedgehog signaling pathways.
Since there are similarities in the mechanisms between silicosis and IPF, and some drugs have been shown to have a role in both silicosis and IPF animal trials (Table 1), it is worth considering the option of obtaining treatment drugs for silicosis from those used for IPF.
Point 3. Section 3.4 is called clinical drugs of silicosis and IPF. Maybe the authors mean approved drugs in China for silicosis and IPF. Because this is a review article for a global audience, it would make sense to broaden the scope and include drugs approved in Europe / USA.
Response 3:
Thanks for your sincere suggestion. In this section on drugs for silicosis, we only address therapeutically approved drugs for clinical use, excluding drugs that are currently in clinical trials. After receiving your advice, we went to relevant websites in Europe and the United States to inquire, except for drugs that have been approved for the symptomatic treatment of silicosis, we have not been able to obtain evidence of clinical approval for the marketing of anti-fibrosis therapy for silicosis. Therefore, we briefly discussed the drugs for symptomatic treatment and marked them in yellow. In addition, in the section on drugs for IPF, we searched the literature and found that currently, the only drugs approved and approved and authorized for the treatment of IPF in Europe, USA, France and other countries are pirfenidone and nintedanib (Cottin V and et al.,2022; Behr, J and et al.,2017).
Ref:
Cottin, V.; Bonniaud, P.; Cadranel, J.; Crestani, B.; Jouneau, S.; Marchand-Adam, S.; Nunes, H.; Wémeau-Stervinou, L.; Bergot, E.; Blanchard, E.; et al. French practical guidelines for the diagnosis and management of idiopathic pulmonary fibrosis - 2021 update. Full-length version. Respiratory medicine and research 2022, 83, 100948, doi:10.1016/j.resmer.2022.100948.
Behr, J.; Günther, A.; Bonella, F.; Geißler, K.; Koschel, D.; Kreuter, M.; Prasse, A.; Schönfeld, N.; Sitter, H.; Müller-Quernheim, J.; et al. German Guideline for Idiopathic Pulmonary Fibrosis - Update on Pharmacological Therapies 2017. Pneumologie (Stuttgart, Germany) 2017, 71, 460-474, doi:10.1055/s-0043-106160.
Revised:
3.4.1. Silicosis
Management of silicosis consists of using bronchodilators and cough medication. However, symptomatic treatment may only ameliorate symptoms rather than restore health. Most drugs that have shown positive effects in animal models, especially reducing lung fibrosis, have not yet been translated into clinically approved drugs in many countries, including Europe and the USA.
